# Real-time gastrointestinal infection surveillance through a cloud-based network of clinical laboratories

**Juliana M. Ruzante**[1]*, **Katherine Olin**[2], **Breda Munoz**[1¤], **Jeff Nawrocki**[2], **Rangaraj Selvarangan**[3], **Lindsay Meyers**[4]

**1** Center for Environmental Health, Risk and Sustainability, RTI International, Research Triangle Park, North Carolina, United States of America, **2** Biomathematics, BioFire Diagnostics, Salt Lake City, Utah, United States of America, **3** Department of Pathology and Laboratory Medicine, Children's Mercy Kansas City, Kansas City, Missouri, United States of America, **4** Medical Data Systems, BioFire Diagnostics, Salt Lake City, Utah, United States of America

¤ Current address: TARGET PharmaSolutions, Durham, North Carolina, United States of America
* jruzante@rti.org

**Data Availability Statement:** There are both legal and ethical restrictions on sharing the data publicly. The data obtained by BioFire is subject to the terms and conditions of a Data Use Agreement (DUA) by

## Abstract

Acute gastrointestinal infection (AGI) represents a significant public health concern. To control and treat AGI, it is critical to quickly and accurately identify its causes. The use of novel multiplex molecular assays for pathogen detection and identification provides a unique opportunity to improve pathogen detection, and better understand risk factors and burden associated with AGI in the community. In this study, de-identified results from BioFire® FilmArray® Gastrointestinal (GI) Panel were obtained from January 01, 2016 to October 31, 2018 through BioFire® Syndromic Trends (Trend), a cloud database. Data was analyzed to describe the occurrence of pathogens causing AGI across United States sites and the relative rankings of pathogens monitored by FoodNet, a CDC surveillance system were compared. During the period of the study, the number of tests performed increased 10-fold and overall, 42.6% were positive for one or more pathogens. Seventy percent of the detections were bacteria, 25% viruses, and 4% parasites. *Clostridium difficile*, enteropathogenic *Escherichia coli* (EPEC) and norovirus were the most frequently detected pathogens. Seasonality was observed for several pathogens including astrovirus, rotavirus, and norovirus, EPEC, and *Campylobacter*. The co-detection rate was 10.2%. Enterotoxigenic *E. coli* (ETEC), *Plesiomonas shigelloides*, enteroaggregative *E. coli* (EAEC), and *Entamoeba histolytica* were detected with another pathogen over 60% of the time, while less than 30% of *C. difficile* and *Cyclospora cayetanensis* were detected with another pathogen. Positive correlations among co-detections were found between *Shigella*/Enteroinvasive *E. coli* with *E. histolytica*, and ETEC with EAEC. Overall, the relative ranking of detections for the eight GI pathogens monitored by FoodNet and BioFire Trend were similar for five of them. AGI data from BioFire Trend is available in near real-time and represents a rich data source for the study of disease burden and GI pathogen circulation in the community, especially for those pathogens not often targeted by surveillance.

and between BioFire and each facility participating in the Trend program. Primarily, the DUA is intended to protect the patient's privacy and comply with applicable privacy laws and regulations. If a data set is requested, BioFire will review such request internally to ensure that any disclosure does not conflict with BioFire's obligations and restrictions set forth in the DUA. For more information, contact Joel Hartsell (joel. hartsell@biofiredx.com).

**Funding:** This study was funded by BioFire Diagnostics. BioFire Diagnostics funded the study at RTI International and provided support in the form of salaries for the following authors: KO, JN, and LM (LM is no longer an employee of BioFire). The specific roles of these authors are articulated in the 'author contributions' section. The funders had no additional role in study design, data collection and analysis, decision to publish, or preparation of the manuscript.

**Competing interests:** The authors have read the journal's policy, and the authors of the manuscript have the following competing interests to declare: At the time that the study was conducted, KO, JN, and LM were all paid employees of BioFire. KO, JN, and LM hold stocks and shares of BioMérieux (owner of BioFire). RS is also on BioFire's Scientific Advisory Board, and has received funding for research, unrelated to the present study, from: BioFire Diagnostics; Cepheid; Abbott; Becton and Dickinson; Hologic; Diasorin; Luminex; Merck; and Bacterioscan. This does not alter our adherence to PLOS ONE policies on sharing data and materials. There are no patents, products in development or marketed products associated with this research to declare.

## Introduction

Every year around the world, 1.5 to 2.5 million children under the age of five die from diarrhea, one of the main symptoms of acute gastrointestinal infections (AGI) [1]. In the United States, AGI accounts for 178.8 million cases a year [2], representing a significant burden to public health. AGI can be caused by a wide range of noninfectious and infectious agents. To effectively control and reduce the disease burden associated with AGI, it is critical to understand which agent is causing the disease, how frequently and when infection occurs in the population, and the potential sources of exposure. Surveillance systems often rely on cases that are reported to public health authorities to compile and analyze this information and monitor the occurrence of illnesses in a population. Among the critical events that must occur for a disease to be reported to authorities is the identification of the AGI agent(s) by clinical laboratories [3]. The quick and accurate identification of the cause of illness is also essential for the effective treatment of AGI patients, which can result in shorter hospital stays, appropriate use of antimicrobials, avoidance of unnecessary isolation of patients, and cost savings [4, 5].

Bacteria, viruses, and parasites are among the infectious agents frequently associated with AGI [6]. Historically, clinical diagnoses and surveillance efforts have been mainly focused on bacteria and relied on culture methods to detect and characterize GI pathogens. Culture is highly specific, only detecting viable organisms that can be cultured. Culture has been the diagnostic pillar for foodborne disease surveillance providing isolates that can be furthered analyzed to obtain information for outbreak detection, food source attribution, and antimicrobial resistance monitoring [7–9]. However, culturing methods can be expensive, time consuming, and limited to organisms for which methods have been validated [4]. Given that culture methods for most pathogens are specific, and often AGI symptoms are generic, it might not always be evident to physicians which type of culture tests to order, potentially resulting in the request of diagnostic tests that will not identify the actual cause of the infection [4, 10, 11]. Novel molecular based culture-independent diagnostics tests (CIDTs), on the other hand, can detect ten or more pathogens in a single test, increasing the likelihood of identifying pathogens especially for which culture is not available or not often requested [4, 5]. Further, CIDTs are highly sensitive, faster, and generally more cost-effective than culture methods [4, 5]. For those reasons, their usage in clinical laboratories is increasing, even though they do not yield an isolate and may detect genetic material from non-viable organisms unrelated to disease.

One example of an FDA-cleared molecular based CIDT, is the BioFire® FilmArray® Gastrointestinal (GI) Panel [12]. The technology used is based on the extraction, amplification, and detection of target nucleic acid sequences by real-time polymerase chain reaction (RT-PCR) and melting curve analysis. The BioFire GI Panel detects 22 pathogens, including 13 bacteria, 5 viruses, and 4 parasites in patients' stool samples [12]. The BioFire GI Panel is a qualitative test that has been validated with human stool collected in Cary Blair transport medium [12]. The indications for use of the test are only for patients with signs and symptoms of AGI and its methodology has been described in detail previously [12, 13]. Clinical laboratories using BioFire® FilmArray® Panels for respiratory and gastrointestinal pathogens have the option to upload de-identified test results to a BioFire cloud database named BioFire® Syndromic Trends (Trend) in near real-time [14, 15]. BioFire Trend then aggregates results from participating clinical laboratories across the globe. The use of BioFire Trend for the surveillance of respiratory pathogens has been previously described [15], but there is no equivalent study conducted on GI pathogens.

While CIDTs pose some immediate challenges to the current way GI pathogens are surveyed by not immediately yielding an isolate [7–9], they also provide a unique opportunity for the public health community to both improve pathogen detection and better understand the

circulation and burden of AGI in the community. To explore and convey the potential use of CIDTs, data from BioFire Trend was obtained, described, and a subset was compared to data from the Foodborne Diseases Active Surveillance Network (FoodNet), an active surveillance system from the Centers for Disease Control and Prevention (CDC) for selected foodborne pathogens frequently associated with AGI [3].

## Materials and methods

BioFire Trend GI data was extracted in July 2019 for the period of January 01, 2016 to October 31, 2018 (34 months). The dataset contained de-identified clinical test variables, anonymized at the participating site level, and included: laboratory location, organism(s) detected, approximate date of the test (obfuscated for privacy) for all participating BioFire Trend sites using the BioFire GI Panel. The test time obfuscation process imports tests from each site in sets of three tests, leading to an average of hours to a couple days between actual time and time recorded in BioFire Trend. To ensure the panel was being routinely used for patient testing and not for test validation, only sites utilizing an average of 10 or more GI tests per month with a median monthly test count within 20% of the mean were included in this study. Additionally, only sites located in the United States were selected. An IRB waiver was obtained as the study did not involve private, identifiable human subjects data, interactions with any human subject, and did not constitute research involving human subjects as defined by the United States Code of Federal Regulations (45 CFR 46.102) [16].

Time series plots were calculated using a centered three-week moving average for the number of tests and number of detections in BioFire Trend. The overall percentage of positives was calculated as the number of pathogen detections over the total number of GI tests performed, including negative tests, for all the 34 months. Further, to estimate seasonality, pathogen specific percentage of positives were calculated overtime using a centered three-week moving average where the number of detections for a given pathogen was divided by the total number of tests performed (including negatives).

The overall co-detection rate was estimated as the number of tests positive for multiple pathogens (e.g.: 2 or more pathogens detected in the same test) divided by the total number of tests, including negative tests. The proportion of co-detection for each pathogen was estimated as the number of a given pathogen detected with one more organism (i.e.: 2 detections) or two (i.e.: 3 detections) divided by the total number of the given pathogen detected during all the 34 months.

To further explore the potential associations between different GI pathogens being detected together, pairwise pathogen correlations were conducted and the phi coefficient was calculated for each pathogen combination across all positive tests with two and three pathogens detected in the same test.

The proportion of detection for eight pathogens monitored by CDC's FoodNet surveillance network: *Salmonella*, *Campylobacter*, *Shigella*, *Yersinia*, Shiga toxin-producing *Escherichia coli* (STEC), *Vibrio*, *Cryptosporidium*, and *Cyclospora* was calculated to compare their magnitude of detection and relative ranking with data obtained from BioFire Trend for the same time period. FoodNet regularly contacts the clinical laboratories located in 10 states, covering about 15% of the United States population [3]. Data from FoodNet is still subjected to underreporting and underdiagnosis [3], however out of the surveillance systems for the eight pathogens above, its data is the most reliable, accessible, and therefore was selected as the dataset for our comparison. The number of confirmed and probable pathogen infections were obtained from FoodNet for 2016 and 2017 via an email request to the surveillance program at CDC. The probable cases included results where CIDT was positive and reflex culture (i.e., culture of

CIDT-positive specimens) was either negative or was not performed. The FoodNet proportion of detection for each of the eight pathogens was calculated by dividing the total pathogen count by the sum of all FoodNet confirmed and suspected pathogens. The same method was applied to the BioFire Trend dataset for these eight pathogens. BioFire Trend reports *Vibrio cholerae* and *Vibrio* non-*cholerae*, as well as STEC and *E. coli* O157, however in the data received from FoodNet those pathogens were only characterized as *Vibrio* and STEC. Therefore, the detections recorded in BioFire Trend for those pathogens were combined respectively into a single *Vibrio* and STEC category to ensure similar comparisons.

All descriptive and statistical analyses were done using Python 3.7.

## Results

BioFire Trend data included a total of 45 clinical laboratories, which consisted of i) laboratories receiving pediatric and non-pediatric patient samples from many medical institutions to one location (20% (9/45)), and ii) in-house laboratories servicing large (>500 beds, 22% (10/45)), medium (100–500 beds, 40% (18/45)), or small hospitals (<100 beds, 18% (8/45)). Out the 45 clinical laboratories, 29 (3 pediatric and 26 general population) met the test utilization criteria for this analysis. From January 01, 2016 through October 31, 2018; 91,401 GI tests were performed by the selected laboratories across 19 states in the United States. Table 1 shows the distribution of tests by year and across the four United States regions as well as the states where the participating laboratories were located. Out of all the tests, 38,943 (42.6%) were positive (detected one or more pathogens) and 52,458 (57.4%) were negative. The total number of tests performed increased approximately 10-fold during the period of the study, from 127 tests per week in January 2016 to 1,226 tests per week in October 2018 (Fig 1). Along with the growth in usage of GI tests, the number of pathogens detected also increased over time (Fig 1). A total of 50,192 organisms were detected during the 34 months of the study. Bacterial pathogens accounted for 71% (35,421/50,192) of all organisms detected, followed by viruses (25%; 12,588/50,192), and parasites (4%; 2,183/50,192). *Clostridium difficile* (30.4%; 15,257/50,192), Enteropathogenic *Escherichia coli* (EPEC) (15.6%; 7,848/50,192), and norovirus (11.1%; 5,567/50,192;), represented over half (57.1%) of the pathogens detected during the 34 months of the study (Fig 2).

**Table 1. Distribution of test number by United States region and year (N = 91,401) for the 29 clinical laboratories.**

| Year | Region | States | # Tests in Trend |
|------|--------|--------|------------------|
| **2016** | West | AK, CA[a], CO[a], HI, UT, WA | 4,511 |
| | South | GA[a], SC, TX | 2,531 |
| | Northeast | NY[a] | 612 |
| | Midwest | IL, IN, KS, NE, OH, SD, WI | 7,353 |
| **2017** | West | AK, CA[a], CO[a], HI, UT, WA | 9,216 |
| | South | GA[a], SC, TX | 3,801 |
| | Northeast | NY[a], PA | 4,154 |
| | Midwest | IL, IN, KS, MI, NE, OH, SD, WI | 14,035 |
| **2018** | West | AK, CA[a], CO[a], HI, ID, UT | 9,865 |
| | South | FL, GA[a], SC, TX | 3,424 |
| | Northeast | NY[a], PA | 9,716 |
| | Midwest | IA, IL, IN, KS, MI, NE, OH, SD, WI | 22,183 |
| **Total** | | | **91,401** |

[a] States (or selected counties in CA and CO) are also FoodNet sites.

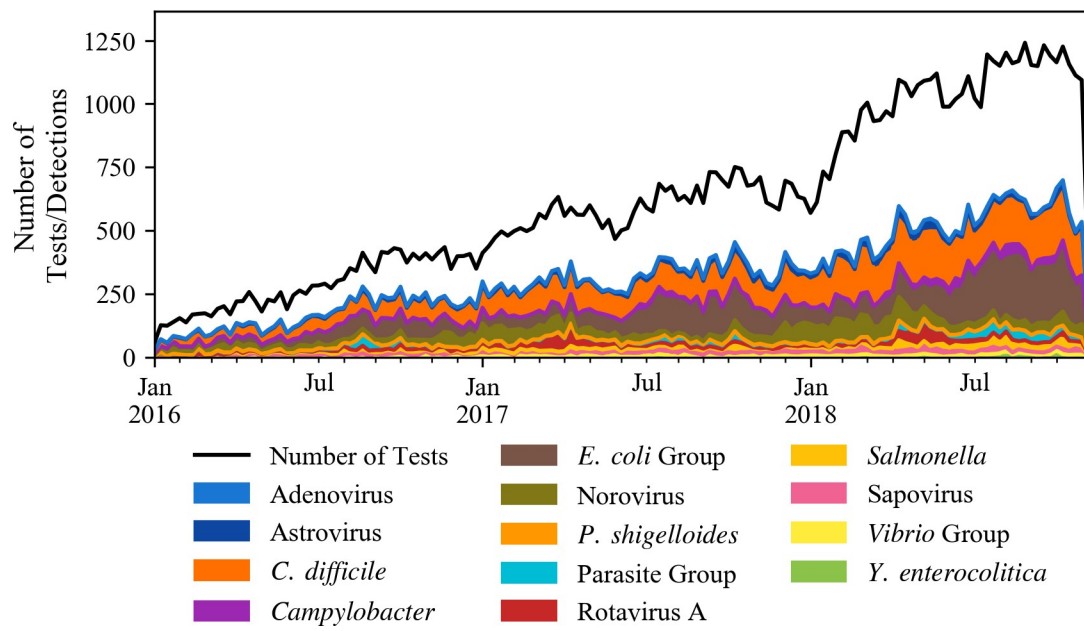

**Fig 1. Number of BioFire Trend GI tests collected from the selected laboratories (N = 91,401) and detected pathogens from January 2016 through October 2018 (N = 50,192).**

The pathogen weekly percentage of positives are shown in Fig 3. Overall, the data from Bio-Fire Trend suggests a clear seasonality for several GI pathogens. Astrovirus and norovirus had increased percentage of positives during the winter months (January–March), while the detection of rotavirus increased in the Spring (April and May). Both sapovirus and adenovirus did

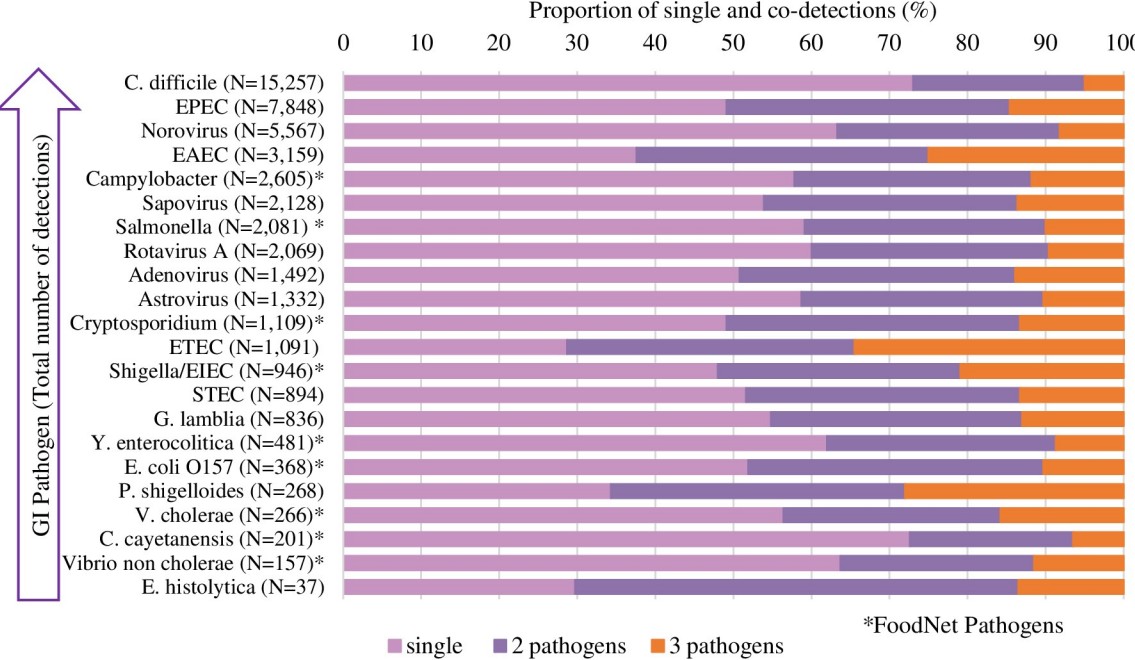

**Fig 2. Total number of pathogen detections and overall proportion of single and co-detections for 22 GI pathogens in BioFire Trend from January 2016 through October 2018 (N = 50,192).**

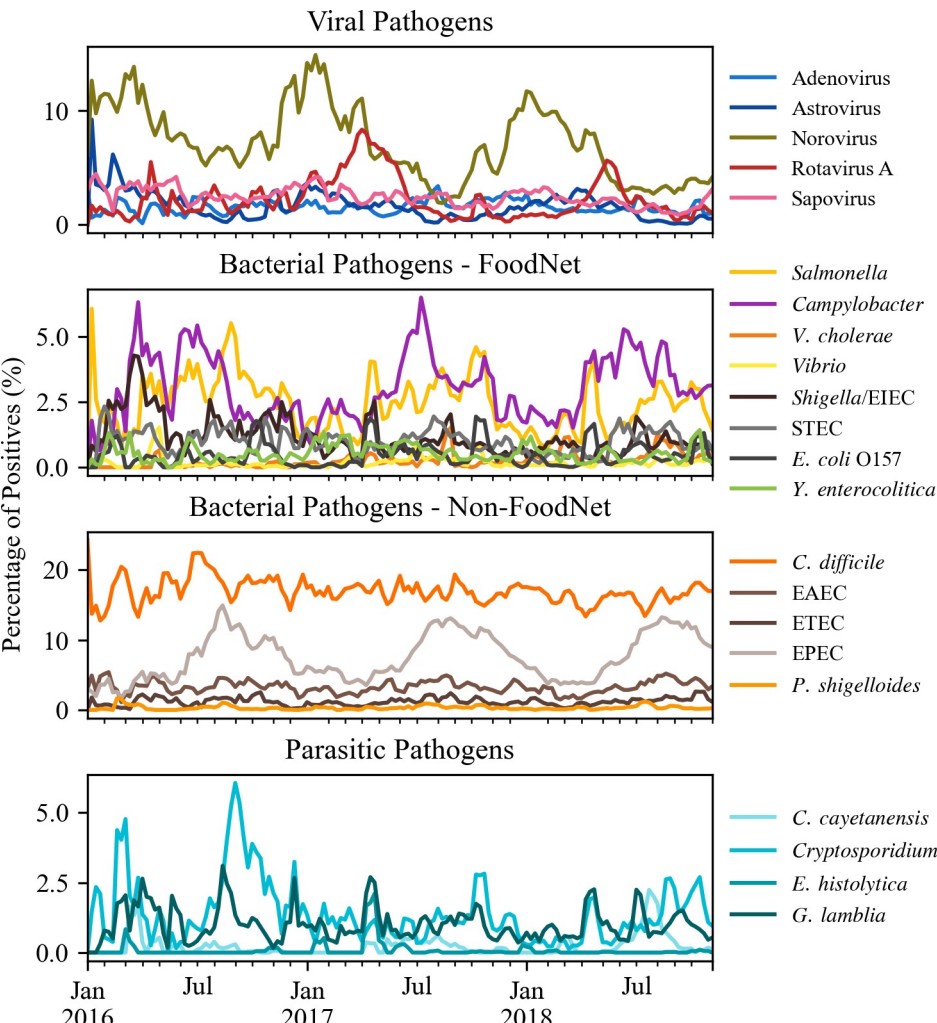

**Fig 3. Weekly percentage of positives for viral (N = 12,588), bacterial FoodNet (N = 7,798), bacterial non-FoodNet (N = 27,623), and parasitic (N = 2,183) GI pathogens detected using BioFire Trend from January 2016 through October 2018.**

not seem to vary in their percentage of positive over the period of the study (Fig 3). No seasonal trends were apparent for *C. difficile*, *Y. enterocolitica*, and the *Vibrios*; but percentage of positives increased during the summer months (June-September) for *Campylobacter*, *Salmonella* (less pronounced), and *Plesiomonas shigelloides*. The percentage of positives of EPEC had clear, distinct peaks during the months of August and September; however, those were not as pronounced for the other *E. coli* types, including *Shigella*/Enteroinvasive *E. coli* (EIEC) among others. Of the parasitic GI pathogens evaluated, a significant increase in the percentage of positives was observed for *Cryptosporidium* around September 2016. Only a few *Cyclospora cayetanensis* detections (N = 201) were observed; however, across all years those were primarily concentrated between the months of March and September, with increased percentages around July. The pattern of detection for *G. lamblia* had several peaks during the 34 months for the study, however no distinct season pattern was observed. Only 37 positives were associated with *Entamoeba histolytica*, with more than 50% (20/37) occurring in October 2016 and April 2017.

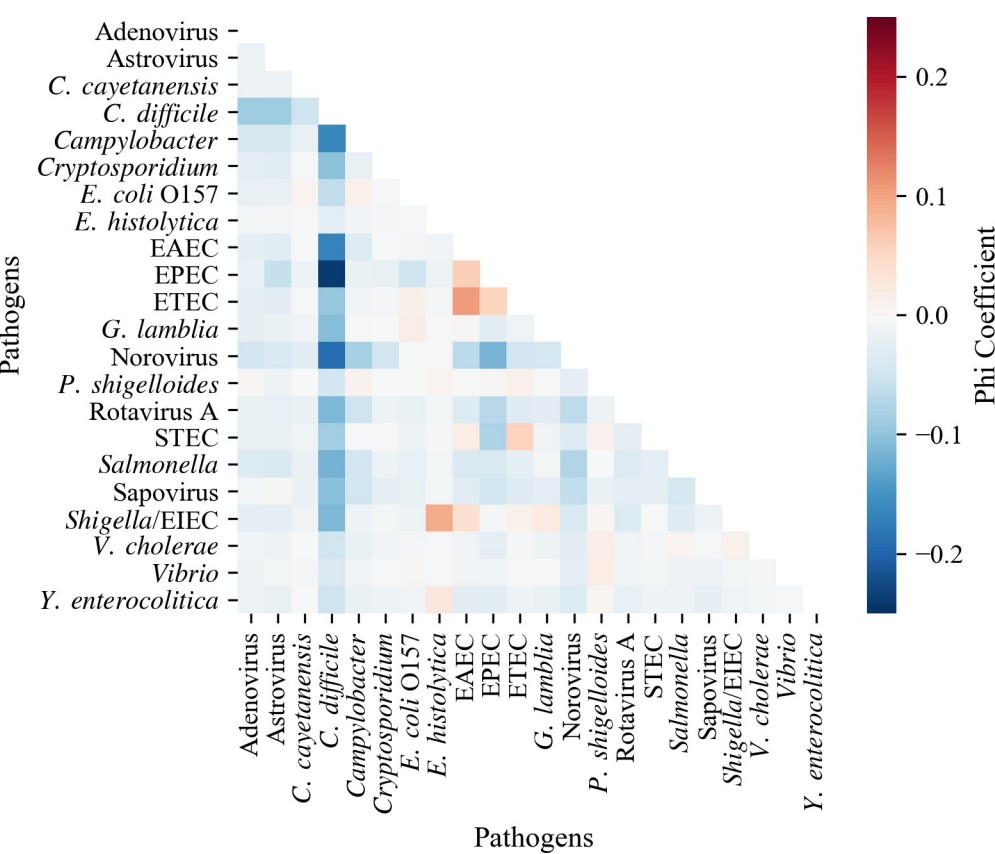

**Fig 4. Pathogen correlation for all positive tests with two or more pathogens detected (N = 9,356).** (Phi coefficient for each pairwise correlation. Darker red indicates stronger positive correlation between two pathogens, while darker blue indicates negative correlation. The scale range is -0.25 to 0.25).

The overall co-detection rate was 10.2% (9,356/91,401) of the total number of tests or 24.0% of positive tests (9,356/38,943). Co-detections with two pathogens (79.8%; 7,463/9,356) were more frequent than co-detections with three organisms (20.2%; 1,893/9,356) detected. In the dataset there were 3,889 tests with four or more pathogens detected, however those were excluded from the analysis due to the likelihood that those were validation tests and not actual patient samples. Fig 2 shows the proportion of single detections as well as co-detections (two or three organisms detected) for each of the pathogens from the BioFire GI Panel. Overall the co-detection proportions varied from 26.9% to 71.3% depending on the pathogen. Over 60% of the detections of ETEC, *E. histolytica*, *P. shigelloides*, and EAEC were co-detections; while less than 30% of the detections of *C. difficile* and *C. cayetanensis*, included another pathogen (Fig 2).

The phi coefficient for each pathogen combination is shown in Fig 4. There was not a strong correlation between any of the pathogen pairs (i.e.: coefficients values between ±0.25); however, *C. difficile* was generally negatively correlated with other pathogens, especially with EPEC (-0.24). Conversely, the Shigella/EIEC-*E. histolytica* and ETEC–EAEC pairs have a weak positive correlation with each other, with phi coefficients equal to 0.10 and 0.11, respectively.

During 2016 through 2017, there were a total of 49,736 confirmed and suspected infections captured by FoodNet for the eight pathogens *Salmonella* (16,159), *Campylobacter* (18,145), *Shigella* (5,076), *Yersinia* (816), STEC (5,039), *Vibrio* (596), *Cryptosporidium* (3,686), and *Cyclospora* (219). The proportion of detection for these eight pathogens were compared in Fig

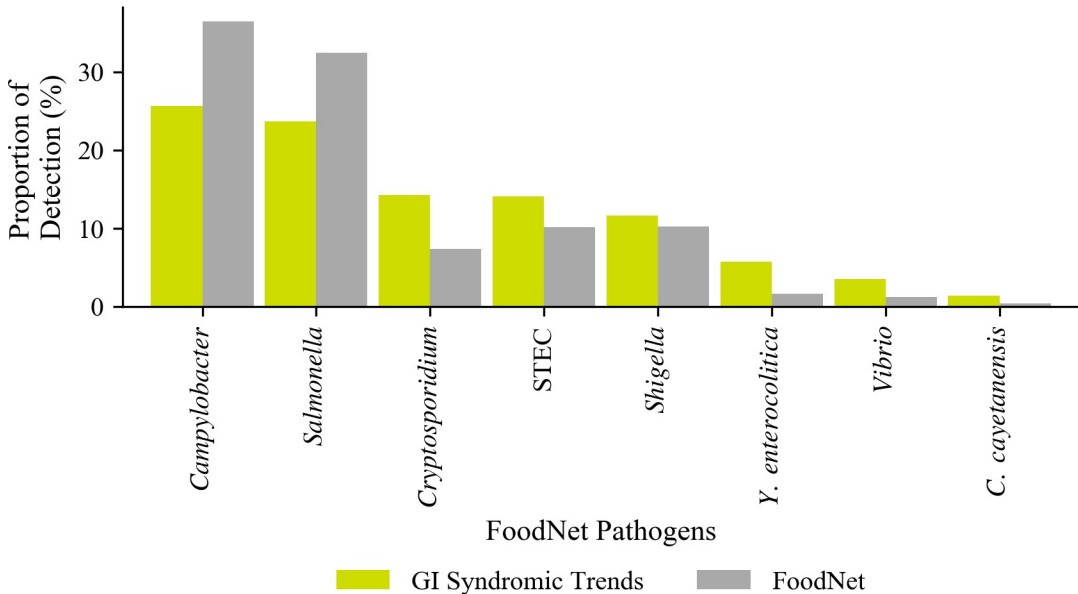

**Fig 5. Proportion of detection and relative ranking of eight pathogens monitored by FoodNet (N = 49,736) and reported to BioFire Trend (N = 4,832) during 2016 and 2017.**

5. Overall, the relative ranking of detections for the eight pathogens were similar between the two data sources, with *Campylobacter* and *Salmonella* being the top pathogens for both datasets during 2016 and 2017, and *Y. enterocolitica*, *Vibrio* and *C. cayetanensis* being the lowest ranking for each. Although the relative rankings were comparable, over 69% of the detections in FoodNet were *Campylobacter* and *Salmonella* while those pathogens only made up 50.2% of BioFire Trend results for those eight pathogens (Fig 5).

## Discussion

This is the first study to summarize and evaluate a large dataset of test results for 22 GI pathogens obtained from 29 clinical laboratories across the United States that use a multiplex real-time PCR assay, the BioFire GI Panel. The results were obtained from BioFire's cloud database, BioFire Trend, Syndromic Trends (Trend), which collects, aggregates, and displays calculated de-identified test results in near real-time for participating sites. BioFire Trend data represents patients who seek medical care in institutions that not only used the BioFire GI Panel, but also participated in BioFire Trend. For those reasons, results are not representative of the United States. Nevertheless, the overall percentage of positives, negatives, co-detection rate, as well as the most frequently detected pathogens (i.e.: *C. difficile*, EPEC, norovirus, EAEC, and *Campylobacter*) reported in this study were similar to what has been reported in previous studies that used the BioFire GI Panel in clinical settings [12, 17–21].

*C. difficile* was the most frequently detected pathogen in this study; however, without further information such as age and medical history of patients, the clinical and public health relevance of this finding is difficult to interpret but is consistent with what others have found when working with this type of dataset [17–21]. Rates of *C. difficile* are also likely to have been overestimated since we were not able to identify and exclude test results from children under 2 years of age; a population known to have high colonization rates of *C. difficile* [22] and for which testing for this pathogen is not generally recommended [23]. EPEC and EAEC were also detected frequently, but not much is known about the domestically acquired burden of those

two pathogens in developed countries [24, 25]. A study conducted in Minnesota suggested that EAEC represents an important GI pathogen and that the proportion of domestically acquired EAEC infections might be underestimated [24]. Further, it has been suggested that AGI can change host inflammation causing an imbalance in the host microbiota leading to an outgrowth of *Enterobacteriales* [17, 25]. This could potentially justify the high numbers of EAEC and EPEC found in this study since over 50% of the detections of these two pathogens were associated with other highly prevalent GI pathogens such as *C. difficile*, norovirus, *Salmonella* and *Campylobacter*. The high frequency of detection of EAEC and EPEC found here suggests that those pathogens have a more significant impact in the burden of AGI than previously thought and requires further research to better characterize their epidemiology as well as sources of exposures and risk factors. Norovirus, on the other hand, has a known burden of illness and is estimated to cause over 21 million AGI annually in the United States, making it the leading cause of foodborne diseases, followed by *Salmonella* [26]. Not surprisingly, norovirus was the third most frequently detected pathogen in this study and also showed a clear seasonality pattern aligned with surveillance reports from the CDC [27].

In addition to norovirus, distinct seasonal trends were observed for rotavirus, astrovirus, and *Campylobacter* for the 34-month study period. These findings are in agreement with data published in the literature and with surveillance reports presented by CDC [28–30]. *P. shigelloides* and EPEC showed distinct peaks in the percentage of positives during the summer months, also consistent with the literature [31, 32]. The peak percentage rates for *P. shigelloides* were particularly pronounced during July. This pathogen is often undetected by culture methods because of its small colony size and/or low prevalence in stool samples. As such, *P. shigelloides* can be easily overlooked; however, PCR-based methods, such as the BioFire FilmArray have increase its detection [32]. A review published on *P. shigelloides* highlighted the need to investigate risk factors associated with both intestinal and systemic infections and the frequency of co-detection of this pathogen with others [32]. The seasonality of *Cryptosporidium*, suggested by the increase in the percentage of positives for the pathogen around September and October, differs from data from the CDC that indicates outbreaks peak during summer months [33]. A possible explanation for this difference is that BioFire Trend is capturing sporadic cases of *Cryptosporidium*, as well as small, localized outbreaks that go undetected by public health agencies given the complexities of disease reporting and potential for underdiagnosis and underreporting [2, 3].

With the increased use of novel molecular CIDTs, the rates of detection of multiple pathogens have also increased. In 2017, the CDC's Food Safety Modernization Act Surveillance Working Group highlighted the need to better understand those types of detections and their significance [34]. However, only a few studies in the United States have been published describing the interactions between GI pathogens. In a study conducted at the University of Washington and Harborview Medical Centers, co-detections represented 9.8% of all detections of GI pathogens [19]. Patients with more than one pathogen detected in their stool, tended to be younger, and more likely to have traveled internationally [19]. In the current study, the overall patterns of co-detection (i.e., rates, proportions, and correlations) for all 22 pathogens detected over 34 months in 29 medical institutions located across the United States were characterized and was found to be similar to previous published studies [12, 18, 19]; however, because de-identified data was used, further characterization was not possible. The correlations found between pathogens were weak, however, this could be potentially helpful in generating hypotheses that could be further investigated. For instance, while *C. difficile* was detected in high numbers during the study period, less than 30% of the detections occurred with other organisms and there were negative correlations between the pathogens detected with *C. difficile* (Fig 4). It might be possible that some of the patients had history of recent

antibiotic treatment thereby pre-disposing them to *C. difficile* while reducing the presence of other GI pathogens [35].

Direct comparisons between Trend and FoodNet rates of detection would not have been appropriate with the data available in this study since they covered different regions in the country (Table 1). However, an initial comparison of the proportion of detection for eight FoodNet pathogens (*Campylobacter*, *Cyclospora*, *Salmonella*, STEC, *Cryptosporidium*, *Shigella*, *Vibrio*, and *Yersinia)* was done to assess, at the higher level, the magnitude and relative ranking of those eight enteric pathogens across the two datasets. Our findings demonstrated that overall, the relative ranking of the eight pathogens is quite similar with exception of *Cryptosporidium* and *Shigella*. Further, the proportion of lower ranking pathogens in BioFire Trend (i.e.: *Yersinia*, *Vibrio* and *Cyclospora*) were higher than in FoodNet potentially due to the comprehensive nature of the BioFire GI Panel (i.e.: every stool is tested for all 22 GI pathogens) which increases the chances of detecting those pathogens that otherwise would not have been identified since clinicians might be less likely to order culture tests for those types of pathogens [11]. A region-by-region comparison between data from BioFire Trend and FoodNet were not possible at the time of the study, because only a few contributing states overlapped (Table 1). Comparisons over time were not done as new participating sites were constantly added during the study period.

In this study, the potential for outbreak detection was not investigated; however, it has been suggested in the literature that the BioFire GI Panel could play a role in early detection of outbreaks [10]. Our data for *C. cayetanensis* also suggests this possibility. In Fig 3, there are two distinct peaks in the percentage of detection for *C. cayetanensis* around July 2016 and July 2017; those coincide with two multistate outbreaks detected by the CDC around the same time [36, 37]. The CDC epidemiologic curve is only available for the 2017 outbreak [36] and also shows a sharp increase in the number of cases in the month of July, suggesting BioFire Trend detected the same pattern. There might be potential to explore outbreak detection for those pathogens with discrete percent of positives peaks and that are detected infrequently.

A cornerstone of BioFire Trend is the protection of patient's privacy. For this reason, the data does not contain any information on age, risk factors, exposure, or medical history. This limitation poses some challenges on how the dataset can be used when interpreting the results, as it has been identified in this paper. However, the benefit of BioFire Trend to public health is not related to the scientific questions that could be answered if demographic and medical histories were available. As demonstrated in this paper, BioFire Trend offers an opportunity to: i) improve the overall understanding of the burden of AGI in the community (especially for those pathogens not often routinely tested or surveyed), ii) track pathogen seasonality, iii) generate scientific hypothesis, and iv) potentially detect outbreaks and monitor pathogens trends over time. Because BioFire Trend results are uploaded to the cloud database within a day, on average, near real-time monitoring of changes in the frequency of detection are possible.

The adoption of multiplex CIDTs is increasing rapidly as shown by the 10-fold increase in tests usage in the 34-month study period, representing a unique chance for public health, as well as the opportunities for the scientific and medical community to learn more about the epidemiology of these pathogens.

## Author Contributions

**Conceptualization:** Juliana M. Ruzante, Rangaraj Selvarangan.

**Data curation:** Juliana M. Ruzante, Katherine Olin, Lindsay Meyers.

**Formal analysis:** Juliana M. Ruzante, Katherine Olin, Breda Munoz, Jeff Nawrocki, Lindsay Meyers.

**Funding acquisition:** Juliana M. Ruzante.

**Investigation:** Juliana M. Ruzante.

**Methodology:** Juliana M. Ruzante, Breda Munoz.

**Project administration:** Juliana M. Ruzante.

**Supervision:** Juliana M. Ruzante, Rangaraj Selvarangan.

**Visualization:** Katherine Olin, Jeff Nawrocki.

**Writing – original draft:** Juliana M. Ruzante.

**Writing – review & editing:** Juliana M. Ruzante, Katherine Olin, Jeff Nawrocki, Rangaraj Selvarangan, Lindsay Meyers.

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
