## [Decision Letter · Decision Letter 0]

7 Jan 2021

PONE-D-20-22634

Real-time gastrointestinal infection surveillance through cloud-based network of clinical laboratories

PLOS ONE

Dear Dr. Ruzante,

Thank you for submitting your manuscript to PLOS ONE. After careful consideration, we feel that it has merit but does not fully meet PLOS ONE’s publication criteria as it currently stands. Therefore, we invite you to submit a revised version of the manuscript that addresses the points raised during the review process.

I have received the review of your paper.  While your paper addresses an interesting question, there are several issues that need to be addressed, please see reviewer’s insightful comments below.  In addition, I have couple points that need to be clarified.  1) can you specify what challenges that CIDTs  (line 84 – 85); 2) how does the results discussed (line 229 – 232) compare to the Foodnet? One minor point, please change “…and melt analysis.” To “…and melting curve analysis.” (line 74).

We look forward to receiving your revised manuscript.

Kind regards,

Baochuan Lin, Ph.D.

Academic Editor

PLOS ONE

Journal Requirements:

2. Please provide a list of the BioFire Trend sites included in this study.

3. In your ethics statement in the Methods section and in the online submission form, please provide additional information about the data used in your retrospective study.

Specifically, please ensure that you have discussed whether all data were fully anonymized before you accessed them and/or whether the IRB or ethics committee waived the requirement for informed consent.

If patients provided informed written consent to have data from their medical records used in research, please include this information.

4. Please include the date(s) on which you accessed the databases or records to obtain the data used in your study.

6. Thank you for providing the following Funding Statement: 

'This study was funded by BioFire Diagnostics (https://www.biofiredx.com/products/filmarray/). Further, the authors: Katherine Olin, Jeff Nawrocki and Lindsay Meyers are paid employees of BioFire Diagnostics and hold shares of BioMérieux (owner of BioFire). Dr. Selvarangan is also on BioFire's Scientific  Advisory Board, and has received funding for research from BioFire Diagnostics, as well as other companies.'

a. We note that one or more of the authors is affiliated with the funding organization, indicating the funder may have had some role in the design, data collection, analysis or preparation of your manuscript for publication; in other words, the funder played an indirect role through the participation of the co-authors.

If the funding organization did not play a role in the study design, data collection and analysis, decision to publish, or preparation of the manuscript and only provided financial support in the form of authors' salaries and/or research materials, please review your statements relating to the author contributions, and ensure you have specifically and accurately indicated the role(s) that these authors had in your study in the Author Contributions section of the online submission form. Please make any necessary amendments directly within this section of the online submission form.  Please also update your Funding Statement to include the following statement: “The funder provided support in the form of salaries for authors [insert relevant initials], but did not have any additional role in the study design, data collection and analysis, decision to publish, or preparation of the manuscript. The specific roles of these authors are articulated in the ‘author contributions’ section.”

If the funding organization did have an additional role, please state and explain that role within your Funding Statement.

Reviewers' comments:

Reviewer's Responses to Questions

**Comments to the Author**

1. Is the manuscript technically sound, and do the data support the conclusions?

Reviewer #1: Yes

2. Has the statistical analysis been performed appropriately and rigorously? 

Reviewer #1: Yes

3. Have the authors made all data underlying the findings in their manuscript fully available?

Reviewer #1: Yes

4. Is the manuscript presented in an intelligible fashion and written in standard English?

Reviewer #1: Yes

5. Review Comments to the Author

Reviewer #1: The paper by Ruzante et al. details their findings on detection and epidemiology of AGI pathogens using de-identified BioFire FilmArray data uploaded to their Trend software over a period of 34 months by 29 clinical laboratories spread across the U.S. The authors present data on the seasonality of AGI pathogens, co-detection rates for each and compare their findings to data available from the CDC FoodNet for a subset of 8 pathogens.

The paper is clearly and concisely written and requires only a few minor modifications.

Line 20: Specify in the Abstract that these are anonymized or de-identified results.

Line 95: For the approximate date of testing, within how many days does the cloud data typically differ from the actual date? Please specify here if you can.

Lines 104-107: This appears to reiterate what is mentioned in lines 101-102. Could you combine these somehow since they both pertain to generation of time series plots.

Line 141: It appears that designation of the first type of lab should occur prior to mention of “reference laboratories”.

Line 293: Change “access” to “assess”.

References 3 and 16: Consider combining these since they both point to the CDC FoodNet page.

References 10 and 27 are the same.

References 38 and 40 could be combined since they both point to Cyclospora outbreaks in 2017.

6. PLOS authors have the option to publish the peer review history of their article (what does this mean?). If published, this will include your full peer review and any attached files.

Reviewer #1: No

---

## [Author Response · Author response to Decision Letter 0]

10 Apr 2021

ANSWERS TO QUESTIONS SUBMITTED ON 03/08/2021

1.Please provide contact information for the data owner. Katherine Olin (Katherine.Olin@biofiredx.com)

2.Please confirm that authors had no special access to the data, and that any qualified researcher can obtain access to the data in the same way the authors obtained it. We confirm that. However, as mentioned in the response to reviewers, a data user agreement (DUA) is required to grant access to the data by a qualified researcher. That is part of the current terms and conditions that BioFire has with each facility participating in the Trend program.

3. Thank you for providing the following information regarding your funding: "Biofire employees, Katherine Olin, Jeff Nawrocki, and Lindsay Meyers (who is no longer a BioFire employee), were part of the data analysis of this study and helped revising the manuscript. We revised the financial/funding statement to reflect that, please see below:“This study was funded by BioFire Diagnostics (https://nam04.safelinks.protection.outlook.com/?url=https%3A%2F%2Fwww.biofiredx.com%2Fproducts%2Ffilmarray%2F&data=04%7C01%7Cjruzante%40rti.org%7Ce089235adf7846c0e3da08d8e2796359%7C2ffc2ede4d4449948082487341fa43fb%7C0%7C0%7C637508358290331728%7CUnknown%7CTWFpbGZsb3d8eyJWIjoiMC4wLjAwMDAiLCJQIjoiV2luMzIiLCJBTiI6Ik1haWwiLCJXVCI6Mn0%3D%7C1000&sdata=ErqXjBfRN%2BX0ZySyZWcPsUwNlNI%2FFkq6Gfn93rqQZOs%3D&reserved=0). Further, the authors: Katherine Olin, Jeff Nawrocki and Lindsay Meyers, under the direction of Dr. Ruzante, contributed to the data analysis and manuscript preparation. Katherine Olin and Jeff Nawrocki are paid employees of BioFire Diagnostics and Lindsay Meyers was a paid BioFire employee during the time of the study. The three of them hold stocks and shares of BioMérieux (owner of BioFire). Dr. Selvarangan is also on BioFire's Scientific Advisory Board, and has received funding for research from BioFire Diagnostics, as well as other companies.” Dr. Ruzante is current a co-PI in a proposal submitted to BioFire Diagnostics and the company paid for her travel expenses from San Diego to Salt Lake City in 2017 when Dr. Ruzante visited BioFire’s headquarter."

Please clarify the following:

a.Is the "funding for research from BioFire Diagnostics" received by Dr. Selvarangan related to the present study? Dr. Selvarangan did not receive any funding for this study. His institution received grant funding for other BioFire studies in which he participated, not this one

b.Please specify which companies are meant by the "other companies" that provided funding research to Dr. Selvarangan, and clarify whether any or all of these funding sources are related to the present study. Dr. Selvarangan also received funding from Cepheid, Abbott, Becton and Dickinson, Hologic, Diasorin, Luminex, Merck and Bacterioscan. Nonetheless, none of those resources were used to conduct the present study.

c.With regards to Dr. Ruzante, are either the "proposal submitted to BioFire Diagnostics" and/or "travel expenses from San Diego to Salt Lake City in 2017" related to the present study? Dr. Ruzante is collaborating with researchers from Ohio State University (OSU) to conduct a new study that will be funded by BioFire. From the time the revisions were submitted to PLOS One to now, this new project was awarded to Ohio State and RTI will be a sub-awardee to OSU. Dr. Ruzante will serve as co-PI, but the new study is not related to the present one. As for the travel expenses in 2017, that trip was made to discuss, in general terms, opportunities for collaborations, and also for Dr. Ruzante to better understand the technology behind BioFire’s FilmArray diagnostic method and Trend.

4.Are there any patents, products in development or marketed products associated with this research to declare? There are no patents, products in development or marketed products associated with this research.

GENERAL COMMENTS AND QUESTIONS

1. Can you specify what challenges that CIDTs (line 84 – 85): We added text to the introduction to address to provide more context to the challenges that CIDTS pose to public health. Please see lines 58-59 and 86.

2. How does the results discussed (line 229 – 232) compare to the Foodnet? A direct comparison of FilmArray and FoodNet results is not appropriate in this case because the geographic locations do not overlap substantially to allow for comparison of rates of detection (see Table 1). However, since FoodNet is an active surveillance system with high quality data for those eight AGI pathogens, we decided that a preliminary comparison would be desirable. Therefore, we calculated and compared the proportions of detection (total pathogen count was divided by the sum of all pathogens detected) for each of the eight pathogens. This way we were able to compare the relative ranking of these pathogens within the two datasets. That, we believe, is a more appropriate approach given the limitation of the two datasets. We added text to the discussion (lines 296 - 297) to make this limitation clearer. 

3. One minor point, please change “…and melt analysis.” To “…and melting curve analysis.” (line 74). We made that change, thank you for the correction.

4. Please provide a list of the BioFire Trend sites included in this study: The data obtained by BioFire is subject to the terms and conditions of a Data Use Agreement (DUA) by and between BioFire and each facility participating in the Trend program. Primarily, the DUA is intended to protect the patient’s privacy. To do so, the test run data for any participating facility is aggregated with the test run data of three or more participating facilities. This protects patients by making it far less likely that any patient run data can be traced to a particular facility and/or individual. Accordingly, we cannot provide details regarding the participating sites, more than what provided in Table 1.

5. In your ethics statement in the Methods section and in the online submission form, please provide additional information about the data used in your retrospective study. Specifically, please ensure that you have discussed whether all data were fully anonymized before you accessed them and/or whether the IRB or ethics committee waived the requirement for informed consent. If patients provided informed written consent to have data from their medical records used in research, please include this information. The data used in this study was de-identified at the participant laboratory level and imported into Trend already anonymized and without any link to patient information. Therefore, the data received by RTI contained no link that could trace the test results to individuals. RTI and BioFire have a DUA in place that detailed how the data would be stored and who would have access to it. In addition, before starting the study, Dr. Ruzante submitted a waiver to RTI’s IRB office, which concluded that: “the study did not involve private, identifiable human subjects data nor interaction with any human subjects. Therefore, while research, it was not considered research with human subjects. This activity does not constitute research involving human subjects as defined by the US Code of Federal Regulations (45 CFR 46.102) and approval of those activities by the RTI IRB was not necessary.” We added a sentence regarding the IRB – lines 101 – 104.

6. Please include the date(s) on which you accessed the databases or records to obtain the data used in your study. The data was extracted from Trend in July 2019 and we added this information to the text as well (line 95).

7. We note that you have indicated that data from this study are available upon request. PLOS only allows data to be available upon request if there are legal or ethical restrictions on sharing data publicly. We will update your Data Availability statement on your behalf to reflect the information you provide. Use .

8. Financial and funding Statement and Biofire employees’ role in the study: Biofire employees, Katherine Olin, Jeff Nawrocki, and Lindsay Meyers (who is no longer a BioFire employee), were part of the data analysis of this study and helped revising the manuscript. We revised the financial/funding statement to reflect that, please see below:

“This study was funded by BioFire Diagnostics (https://www.biofiredx.com/products/filmarray/). Further, the authors: Katherine Olin, Jeff Nawrocki and Lindsay Meyers, under the direction of Dr. Ruzante, contributed to the data analysis and manuscript preparation. Katherine Olin and Jeff Nawrocki are paid employees of BioFire Diagnostics and Lindsay Meyers was a paid BioFire employee during the time of the study. The three of them hold stocks and shares of BioMérieux (owner of BioFire). Dr. Selvarangan is also on BioFire's Scientific Advisory Board, and has received funding for research from BioFire Diagnostics, as well as other companies.” Dr. Ruzante is current a co-PI in a proposal submitted to BioFire Diagnostics and the company paid for her travel expenses from San Diego to Salt Lake City in 2017 when Dr. Ruzante visited BioFire’s headquarter. The disclosures by the authors in this study do not alter their adherence to PLOS ONE policies on sharing data and materials.

9. Please also provide an updated Competing Interests Statement declaring this commercial affiliation along with any other relevant declarations relating to employment, consultancy, patents, products in development, or marketed products, etc. Please see revised financial statement above (#8) for the conflict-of-interest disclosures. "This does not alter our adherence to PLOS ONE policies on sharing data and materials.”

REVIEWER’S COMMENTS

1. Line 20: Specify in the Abstract that these are anonymized or de-identified results. We added text to clarify this point as well as other instances in the paper. A few minor changes were made to the abstract to meet the word limit.

2. Line 95: For the approximate date of testing, within how many days does the cloud data typically differ from the actual date? Please specify here if you can. The test time obfuscation process imports tests from each site in sets of three tests, leading to an average of hours to a couple days between actual time and time reported in BioFire Trend. Text was added to the manuscript to clarify this point (lines 99 – 101).

3. Lines 104-107: This appears to reiterate what is mentioned in lines 101-102. Could you combine these somehow since they both pertain to generation of time series plots. We made a few changes to that paragraph, but believe it is important to keep the distinction between methods, since they are all slightly different. We hope it is acceptable to this reviewer. 

4. Line 141: It appears that designation of the first type of lab should occur prior to mention of “reference laboratories”. There were no reference labs in this study. We edited for clarity. 

5. Line 293: Change “access” to “assess”. Done, thank you for catching the misspell. 

6. References 3 and 16: Consider combining these since they both point to the CDC FoodNet page. We decided to use the general reference and delete the other. Thank you for catching the duplicate.

7. References 10 and 27 are the same. I believe it is reference 10 and 37, we removed the duplicate. Thank you for catching the duplicate.

8. References 38 and 40 could be combined since they both point to Cyclospora outbreaks in 2017. Those were two different outbreaks, but we could use the general page.

---

## [Editor Report · Decision Letter 1]

14 Apr 2021

Real-time gastrointestinal infection surveillance through cloud-based network of clinical laboratories

PONE-D-20-22634R1

Dear Dr. Ruzante,

We’re pleased to inform you that your manuscript has been judged scientifically suitable for publication and will be formally accepted for publication once it meets all outstanding technical requirements.

Kind regards,

Baochuan Lin, Ph.D.

Academic Editor

PLOS ONE
---

## [Editor Report · Acceptance letter]

22 Apr 2021

PONE-D-20-22634R1 

Real-time gastrointestinal infection surveillance through a cloud-based network of clinical laboratories 

Dear Dr. Ruzante:

I'm pleased to inform you that your manuscript has been deemed suitable for publication in PLOS ONE. Congratulations! Your manuscript is now with our production department. 

Kind regards, 

on behalf of

Dr. Baochuan Lin 

Academic Editor

PLOS ONE